# The Role of Artificial Intelligence in Early Cancer Diagnosis

**DOI:** 10.3390/cancers14061524

**Published:** 2022-03-16

**Authors:** Benjamin Hunter, Sumeet Hindocha, Richard W. Lee

**Affiliations:** 1Early Diagnosis and Detection Centre, The Royal Marsden NHS Foundation Trust, Chelsea, London SW3 6JJ, UK; benjamin.hunter@rmh.nhs.uk (B.H.); sumeet.hindocha@rmh.nhs.uk (S.H.); 2Department of Surgery and Cancer, Imperial College London, London SW7 2BX, UK; 3Artificial Intelligence for Healthcare Centre for Doctoral Training, Imperial College London, London SW7 2BX, UK; 4National Heart and Lung Institute, Imperial College London, London SW7 2BX, UK; 5Early Diagnosis and Detection, Genetics and Epidemiology, The Institute for Cancer Research, London SW7 3RP, UK

**Keywords:** early diagnosis, artificial intelligence, machine learning, deep learning, screening

## Abstract

**Simple Summary:**

Diagnosing cancer at an early stage increases the chance of performing effective treatment in many tumour groups. Key approaches include screening patients who are at risk but have no symptoms, and rapidly and appropriately investigating those who do. Machine learning, whereby computers learn complex data patterns to make predictions, has the potential to revolutionise early cancer diagnosis. Here, we provide an overview of how such algorithms can assist doctors through analyses of routine health records, medical images, biopsy samples and blood tests to improve risk stratification and early diagnosis. Such tools will be increasingly utilised in the coming years.

**Abstract:**

Improving the proportion of patients diagnosed with early-stage cancer is a key priority of the World Health Organisation. In many tumour groups, screening programmes have led to improvements in survival, but patient selection and risk stratification are key challenges. In addition, there are concerns about limited diagnostic workforces, particularly in light of the COVID-19 pandemic, placing a strain on pathology and radiology services. In this review, we discuss how artificial intelligence algorithms could assist clinicians in (1) screening asymptomatic patients at risk of cancer, (2) investigating and triaging symptomatic patients, and (3) more effectively diagnosing cancer recurrence. We provide an overview of the main artificial intelligence approaches, including historical models such as logistic regression, as well as deep learning and neural networks, and highlight their early diagnosis applications. Many data types are suitable for computational analysis, including electronic healthcare records, diagnostic images, pathology slides and peripheral blood, and we provide examples of how these data can be utilised to diagnose cancer. We also discuss the potential clinical implications for artificial intelligence algorithms, including an overview of models currently used in clinical practice. Finally, we discuss the potential limitations and pitfalls, including ethical concerns, resource demands, data security and reporting standards.

## 1. Introduction

Early cancer diagnosis and artificial intelligence (AI) are rapidly evolving fields with important areas of convergence. In the United Kingdom, national registry data suggest that cancer stage is closely correlated with 1-year cancer mortality, with incremental declines in outcome per stage increase for some subtypes [1]. Using lung cancer as an example, 5-year survival rates following resection of stage I disease are in the range of 70–90%; however, rates overall are currently 19% for women and 13.8% for men [2]. In 2018, the proportion of patients diagnosed with early-stage (I or II) cancer in England was 44.3%, with proportions lower than 30% for lung, gastric, pancreatic, oesophageal and oropharyngeal cancers [3]. A national priority to improve early diagnosis rates to 75% by 2028 was outlined in the National Health Service (NHS) long-term plan [4]. Internationally, early diagnosis is recognised as a key priority by a number of organisations, including the World Health Organisation (WHO) and the International Alliance for Cancer Early Detection (ACED). 

Many studies indicate that screening can improve early cancer detection and mortality, but even in disease groups with established screening programmes such as breast cancer, there are ongoing debates surrounding patient selection and risk–benefit trade-offs, and concerns have been raised about a perceived ‘one size fits all’ approach incongruous with the aims of personalised medicine [5,6,7]. Patient selection and risk stratification are key challenges for screening programmes. AI algorithms, which can process vast amounts of multi-modal data to identify otherwise difficult-to-detect signals, may have a role in improving this process in the near future [8,9,10]. Moreover, AI has the potential to directly facilitate cancer diagnosis by triggering investigation or referral in screened individuals according to clinical parameters, and automating clinical workflows where capacity is limited [11]. In this review, we discuss the potential applications of AI for early cancer diagnosis in symptomatic and asymptomatic patients, focussing on the types of data that can be used and the clinical areas most likely to see impacts in the near future. 

## 2. An Overview of Artificial Intelligence in Oncology

### 2.1. Definitions and Model Architectures

AI is an umbrella term describing the mimicking of human intelligence by computers (Figure 1). Machine learning (ML), a subdivision of AI, refers to training computer algorithms to make predictions based on experience, and can be broadly divided into supervised (where the computer is allowed to see the outcome data) or unsupervised (no outcome data are provided) learning. Both approaches look for data patterns to allow outcome predictions, such as the presence or absence of cancer, survival rates or risk groups. When analysing unstructured clinical data, an often-utilised technique, both in oncology and more broadly, is natural language processing (NLP) [12]. NLP transforms unstructured free-text into a computer-analysable format, allowing the automation of resource-intensive tasks.

It is common practice in ML to split data into partitions, so that models are developed and optimised on training and validation subsets, but evaluated on an unseen test set to avoid over-optimism. A summary of commonly used supervised learning methods is provided in Table 1. Such methods include traditional statistical models such as logistic regression (LR) as well as novel decision tree and DL algorithms. 

Deep learning (DL) is a subgroup of ML, whereby complex architectures analogous to the interconnected neurons of the human brain are constructed. Popular Python-based frameworks for deep learning include Tensorflow (Google) and PyTorch (Facebook), which provide features for model development, training and evaluation. Google also provides a free online notebook environment, Google Colaboratory, allowing cloud-based Python use and access to graphic processing units (GPUs) without local software installation. 

Although a detailed description of neural network structures is beyond the scope of this article, artificial neural networks (ANNs) can be used to illustrate the overarching principles (Figure 2). As a recent example, Muhammad et al. used an ANN to predict pancreatic cancer risk using clinical parameters such as age, smoking status, alcohol use and ethnicity [18]. In their most basic form, ANNs consist of: (1) an input layer, (2) a ‘hidden layer’, consisting of multiple nodes which multiply the input by weights and add a bias value, and (3) the output layer, passing the weighted sum of hidden layer nodes to an activation function to make predictions. Deep learning simply refers to networks with more than one hidden layer. 

Many early diagnosis models have exploited convolutional neural network (CNN) architectures, which led to a revolution in computer-vision research by allowing the use of colour images as input data. While the downstream fully connected layers resemble those of an ANN, the input data are processed by a series of kernels which slide over image colour channels and extract features, such as edges and colour gradients. These inputs are then pooled and flattened before being passed to the fully connected layer. Many pre-defined CNN architectures with varying degrees of complexity are available for use, including AlexNet [20], EfficientNet [21], InceptionNet [22], ResNet [23] and DenseNet [24]. As we discuss further in this article, CNNs have a wide range of applications in radiology and digital pathology. 

### 2.2. Data Types: Electronic Healthcare Records

A number of emerging healthcare data modalities are suitable for analysis with AI. In recent years, a global expansion in electronic healthcare record (EHR) infrastructures has occurred, enabling vast amounts of clinical data to be stored and accessed efficiently [25]. Many exciting digital collaborations are arising to facilitate early diagnosis research using EHRs, including the UK-wide DATA-CAN hub [26]. Other digital databases record outcome measures and pathway data. For example, the Digital Cancer Waiting Times Database aims to improve cancer referral pathways through user-uploaded performance metrics [27]. 

It is important to draw a distinction between local hospital EHR data and national public health data registries, including those utilised by multi-centre screening studies. With registries, unified database structures are being implemented for consistency across institutions. A key aim of the NHSx ‘digital transformation of screening’ programme is to ensure interoperability of systems, so that data can flow seamlessly along the entire screening pathway, including into national registry databases [28]. An example of database unification is the new U.K. cervical cancer screening management system, which will simplify 84 different databases into a single national database, and aims to streamline data entry and provide simple, cloud-based access for users [29]. 

Digital databases, whether local or national, are ripe for analysis with AI, which is inherently able to process large amounts of information (‘Big Data’) [30]. EHR data typically include structured, easily quantifiable data such as admission dates or blood results, and unstructured free-text such as clinical notes or diagnostic reports. The latter can be analysed using NLP approaches. An overview of NLP in oncology is provided by Yim et al. [12], and example early diagnosis uses include identifying abnormal cancer screening results [31], auditing colonoscopy or cystoscopy standards [32,33] and identifying or risk-stratifying pre-malignant lesions [34,35,36,37,38]. NLP has also been used to automate patient identification for clinical trials, reducing the burden of eligibility checks [39]. Morin and colleagues published an exciting example of how AI and NLP technology can integrate into EHR systems: their model can analyse millions of data points and perform real-time cancer prognostication based on continuous learning of routinely collected clinical data [40].

### 2.3. Data Types: Radiology

The migration from radiographic film to digital scans within Patient Archive and Communication Systems (PACS) has yielded similar benefits for imaging research. Radiomics refers to quantitative methods for analysing radiology images (including CT, nuclear medicine, MRI and ultrasound scans), and may be divided into traditional ML and DL approaches. For traditional ML approaches, textural features are captured from highlighted regions of interest (ROIs), and relate broadly to size and shape, intensity and heterogeneity readouts. These features are used to train models for classification or prognostication. In the early cancer diagnosis setting, this includes classification of indeterminate nodules or cysts as benign or malignant. Many studies have employed a radiomics approach to accurately classify lung nodules in this fashion [41,42], and Shakir et al. generated accurate radiomics-based cancer likelihood functions across many tumour groups, including lung, colorectal and head and neck cancers [43]. There is also potential to predict indolence versus aggressive disease, which can be a relevant determinant of when early diagnosis is most likely to be of patient benefit. As an example, in 2019, Lu et al. published a four-feature radiomics signature which predicted survival and treatment response in ovarian cancer [44]. 

As discussed above, CNNs are the cornerstone of DL-based medical-imaging classification. If we take the EfficientNet architectures developed in 2019 as an example, they have been successfully applied to diagnosing many cancer types, including breast cancer (AUC 0.95) [19], lung cancer (AUC 0.93) [45] and brain cancer (accuracy 98%) [46] with high performance, while increasing computational efficiency compared with historic models [21]. In addition to benign/malignant classification tasks, many CNN architectures also exist for lesion identification and segmentation, such as U-Net [47] and V-Net [48] models. Such models can be evaluated using the Dice similarity co-efficient (Dice-score), which assesses the degree of overlap between two segmentation masks. For example, Baccouche et al. developed a U-Net model for mammographic breast-lesion segmentation with a Dice score of 96% [49].

The possible benefits and drawbacks of traditional ML and DL approaches are presented in Table 2. A cited advantage of traditional ML models is explainability–features are hand-crafted and defined upfront, and their expression levels can be readily quantified [50]. In contrast, DL has been criticised as a black box, due to the perception that the inner workings are opaque. This criticism becomes less relevant as the field advances, and with the caveats that DL models are computationally more intensive and data-hungry, they widely outperform traditional models in classification and prediction tasks, and may become the dominant force in the near future [51,52]. Hybrid models incorporating both hand-crafted approaches and DL are also arising [53]. 

### 2.4. Data Types: Digital Pathology

Digital pathology, referring to the creation and analysis of digital images from scanned pathology slides, is another important field of AI research relevant to early diagnosis [54]. In a U.K. survey, 60% of institutions had access to digital pathology scanners in 2018, and the global uptake is likely to increase [55]. Schüffler and colleagues’ experience with 288,903 digital slides over a 3-year period demonstrates the power of this technology to improve diagnostic workflows and facilitate large-scale sharing of research data [56]. Many studies require pathological reviews of diagnostic specimens for eligible patients; thus, the use of digital slides has removed previous bottlenecks associated with glass slide transfer and processing, particularly for patients eligible for multiple studies [56]. The authors also describe the benefits of integrated digital programmes, whereby histopathology data are automatically linked with relevant tests, such as molecular results, and viewed on an integrated platform, reducing the inefficiency of opening multiple windows per case [56]. As described by the digital pathology centre of excellence PathLAKE, the COVID-19 pandemic has highlighted many benefits of digital working, include increased work-force resilience, time efficiency savings, outsourcing and easy access to expert supervision and training [57]. The crosstalk between digital pathology and other electronic healthcare systems is an area of international focus highlighted by integrating health-care enterprises (IHEs) [58]. 

CNNs have been widely utilised for cancer detection using automated whole-slide analysis: a model published by Coudray et al. diagnosed lung cancer with an AUC of 0.97 [59], and high diagnostic accuracy has been observed amongst other tumour subtypes [60,61,62]. CNNs are able to perform tumour sub-typing, including the identification of molecular phenotypes and targetable receptors [63,64], and many models have been trained to automate grade and stage assessments [65,66,67,68]. Applications such as Paige-AI could provide clinically available tools for automated analysis, in this case of prostate biopsies based on a CNN model [69]. 

As novel pathology techniques emerge, AI may have a role in processing the more complex data they yield. Multiplex immunohistochemistry, for example, enables the evaluation of multiple cellular subsets on a single pathology slide using unique chromogen labels. Such an approach has enabled detailed analysis of the cancer immune landscape in some subsites [70]. Fassler et al. developed a U-Net-based model to reliably detect and classify six cell populations associated with pancreatic adenocarcinoma [71]. 

Exciting gains have also been made in predictive biomarker analysis. ML models have been used to identify predictive signatures from peripheral blood samples and tumour biopsy material, including analyses of whole-genome profiles [72,73,74].

### 2.5. Data Types: Multi-Omic Data

Given the complexity of tumour biology, models based on single data types could miss important predictive information arising from the interaction between interdependent biological systems. There is, therefore, a drive to integrate multi-model data, which may include radiomic, genomic, transcriptomic, metabolomic and clinical factors, to better describe the tumour landscape and improve diagnostic precision. Several large-scale databases, including ‘LinkedOmics’, which contains multi-omic data for 11,158 patients across 32 cancer types, are available to facilitate the detection of associations between data modalities and assist model development [75]. 

Using central nervous system (CNS) tumours as an example, multi-omic data, including single-nucleotide polymorphism (SNP) mutations (e.g., TARDBP), gene methylation (e.g., 64-MMP) and transcriptome abnormalities (e.g., miRNA-21), are known to predict the progression of meningiomas [76]. A systematic review of multi-omic glioblastoma studies by Takahashi et al. found that most utilised ML techniques for analysis, likely due to the size and complexity of the data [77]. In one study of 156 patients with oligodendrogliomas, mRNA expression arrays, microRNA sequencing and DNA methylation arrays were analysed using a multi-omics approach to better classify 1p/19q co-deleted tumours [78]. Use of unsupervised clustering techniques identified previously undescribed molecular heterogeneity in this group, revealing three distinct subgroups of patients [78]. These subgroups had differences in important histological factors (microvascular proliferation and necrosis), genetic factors (cell-cycle gene mutations) and clinical factors (age and survival) [78]. Franco et al. explored DL autoencoder models to predict cancer subtypes from multi-omic data, included methylation, RNA and miRNA sequencing readouts from The Cancer Genome Atlas (TCGA) [79]. The authors identified three GBM subtypes with differentially expressed genes relating to synaptic function and vesicle-mediated transport [79]. 

These studies demonstrate how machine learning approaches applied to multi-omic data can reveal previously hidden elements of tumour biology, which may have important implications for diagnosis and prognostication. 

## 3. Clinical Applications

Below, we discuss the areas where AI is likely to have clinical impact in the near future, using exemplar cancer groups (Figure 3). 

### 3.1. Risk-Stratified Screening of Asymptomatic Patients

Several large-scale studies have shown that lung-cancer screening in at-risk patients confers survival benefits [80,81]. Subsequently, in the U.S., the Centers for Medicare & Medicaid Services (CMS) deemed that patients aged 55–77 with a ≥30 pack-year smoking history are eligible for CT screening, with new guidelines suggesting that this should be relaxed further [82,83]. However, in practice, only a small proportion of eligible patients are actually screened, partially due to poor smoking-status documentation and physician time pressures [84,85]. To improve screening selection, Lu et al. developed a CNN model incorporating chest X-rays and minimal EHR data (age, sex, current smoking status) to predict 12-year incident cancer risk, which was compared with the CMS criteria [86]. The imaging component was trained using an Inception V4 network on 85,748 radiographs from the PLCO trial and validated in 5615 and 5493 radiographs from the PLCO and NLST studies, respectively. The team found that the model improved upon CMS eligibility criteria, reporting an AUC of 0.755 compared with 0.634, and achieved parity with more complex risk scores requiring 11 data points (PLCO_M2012_) [86]. 

More recently, Gould et al. published an ML model based on non-imaging EHR data [87]. Using a dataset of 6505 patients with lung cancer and 189,597 controls, the model was more accurate than the PLCO criteria at predicting lung cancer within the next 9–12 months (AUC 0.86). Moreover, it improved upon standard eligibility criteria for lung cancer screening, providing evidence that AI-enhanced assessment of routine clinical data can help identify patients for targeted screening programs. Use of AI to improve patient selection for screening may be a useful path to early diagnosis in the future. 

### 3.2. Symptomatic Patient Triage

General practitioners (GPs) are often the first port of call for patients with cancer symptoms, and have a critical role to play as gatekeepers to secondary care [88]. Over the last decade, a number of decision-support tools have emerged to assist GPs in determining which cancer symptoms require referral for further investigation [89]. For example, the CE-marked decision support tool, ‘C The Signs’, is currently being piloted across a number of practices to assist GPs in cancer risk stratification [90,91]. The tool provides a dashboard for use in real time and suggests investigations or referrals based on cancer-symptom profiles. Early evaluation reports suggest an increased cancer-detection rate of 6.4% [91]. It should be noted there are currently no peer-reviewed publications relating to this tool in the literature, and although marketing indicates the utilisation of AI to map decisions to the latest evidence, it is not possible to fully critique its infrastructure without published methodology.

Technologies are also emerging to diagnose and triage patients directly according to self-described symptoms, using chat-bots or online symptom checkers. The commercial digital healthcare provider, Babylon Health, provides patients access to private consultations by phone or computer apps [92]. Babylon utilises a Bayesian network based on disease probability profiles informed by epidemiological data and expert opinion to diagnose diseases and recommend actions, such as attending Accident and Emergency, or booking a non-urgent GP appointment, according to patient-entered symptoms [93]. Its triage and diagnostic system are reported as having comparable accuracy and safety to human clinicians, with the caveat that the use of simulated consultations limits the external validity of this evaluation [93]. It is again important to note that this tool has been implemented clinically despite a paucity of peer-reviewed publications detailing robust testing and validation procedures, which has drawn criticism from the MHRA and oncology community [94].

The current literature suggests that AI may play a role in triaging symptomatic patients in the community at risk of cancer in the future; however, further evidence, including robust prospective validation studies, is needed to confirm their efficacy and safety for clinical deployment.

### 3.3. Diagnostic Workflow Triage

Given increasing concerns about the limited diagnostic workforce and infrastructure, particularly after the COVID-19 pandemic which disrupted diagnostic workflows and halted screening programs [95,96], we are likely to see an increasing role for AI-based workflow triage in the near future. Such systems are intended to screen diagnostic test results and allocate cases for specialist review, for example by pathologists or radiologists, based on risk, so that the large volume of normal or low-risk examinations are not escalated (Figure 4).

A recent paper by Gehrung et al. utilised deep learning to triage pathology workflows [98]. Barrett’s oesophagus (BE), referring to reflux-induced epithelial metaplasia, is a risk factor for oesophageal cancer which requires significant diagnostic resources for surveillance endoscopies and biopsies [99]. The emergence of innovative non-endoscopic approaches, such as Cytosponge, improve the patient experience but exacerbate the pathology resource problem, due to the amount of generated cellular material requiring pathologist review [100]. The team trained a selection of CNN architectures to perform Cytosponge slide quality control and BE detection, and generated a priority system for manual review based on a combination of the type of findings (positive or negative) and model confidence [98]. Five of the eight diagnosis-confidence categories could be fully automated by the chosen CNN while maintaining comparable diagnostic accuracy to a pathologist (sensitivity and specificity 82.5% and 92.7%, respectively). The model was externally validated on 3038 slides from 1519 patients, with a simulated reduction in pathologist workload of 57.2% [98].

In breast cancer imaging, AI can detect mammographic abnormalities with comparable accuracy to radiologists, and a wealth of commercial software packages have come onto market in recent years [101,102,103,104]. A 2020 study by Dembrower et al. evaluated the ability of AI-enhanced triage to reduce radiologist workloads using a set of over 1 million mammograms from 500,000 women [8]. The team tested rule-out thresholds based on AI malignancy-risk scores, and found that most women could be safely triaged to no radiologist review for predicted risks of less than 60% [8]. An enhanced-assessment algorithm was also developed, whereby the AI system gave a second read of mammograms reported negative by radiologists. The team set rule-in thresholds for recommending further evaluation with MRI, and found that for the top 1% of risk scores, 12% and 14% of patients developed interval or screen-detected cancers, respectively [8]. More recently, Yi et al. published performance metrics for DeepCAT, another mammography triage system trained on 1878 images. In the test set of 595 images, the model triaged 315 scans (53%) as low priority [105]. None of the low-priority images contained cancer, again supporting the notion that AI can provide a safe and effective triage of mammograms.

These studies provide good evidence that AI systems can be well integrated into clinical workstreams, and that with appropriate risk thresholding, can reduce the burden of diagnostic work through enhanced triage.

### 3.4. Early Detection

Automating the detection and classification of pre-malignant lesions and early cancers is an area where AI is well established. For image-based models, indeterminate pulmonary nodules are a good candidate, because such nodules are found frequently and are usually benign, with a small proportion representing early-stage cancers [80,81].

Ardila and co-authors at Google published an end-to-end solution, meaning that nodule identification and classification were integrated into one workstream, trained on 42,290 CT scans from 14,851 patients enrolled in the National Lung Screening Trial [52,80]. An example end-to-end pipeline is shown in Figure 5. A region-based convolutional neural network (R-CNN) was developed to register longitudinal scans where available, and whole-CT scan data and bounding-box nodule ROIs were utilised to predict malignancy using a 3D Inception model [52]. The model outperformed the average radiologist at malignancy risk-prediction, and achieved a cutting-edge AUC of 95.5% at external validation in 1139 cases [52]. The model has not been prospectively validated, but could become available for clinical use in the future. 

The lung health company Optellum have developed virtual nodule clinical software based on a lung cancer prediction CNN (LCP-CNN). The IDEAL study is a two-phase study aiming to validate the LCP-CNN retrospectively and prospectively across three healthcare trusts [106]. While the prospective results are awaited, Baldwin et al. report an AUC of 89.6% for malignancy prediction, which outperforms a commonly used risk score (Brock), with a reduction in false negatives, in retrospective evaluation [51]. It is likely that the use of digital nodule-management tools for identifying and risk-stratifying nodules will become commonplace in the next five years, although validation in different patient cohorts will be required. For example, the DART study will evaluate the role of the LCP-CNN in screen-detected nodules.

In addition to X-ray and CT modalities, early detection models trained on bi- or multi-parametric MRI scans are also emerging. In one retrospective study by Schelb et al., a U-Net model was developed using T2 and diffusion-weighted MRI images from 312 patients undergoing evaluation for prostate cancer [107]. The architecture provides probabilities of each voxel (3D pixel) belonging to normal or abnormal prostate tissue and creates an automated segmentation map of the relevant regions. The automated model achieved comparable performance to clinical assessment of prostate lesions (sensitivity 88% vs. 92%, specificity 50% vs. 47%, respectively), and was highly accurate at whole-prostate segmentation (Dice score 0.89) [107]. Many other studies support the conclusion that AI-based early detection algorithms can achieve parity with clinical assessments of prostate MRI, and again, commercial solutions are now available [108,109]. 

Aside from image-based approaches, there is increasing interest in the use of peripheral blood biomarkers for early cancer detection, in part due to easier access to next-generation sequencing (NGS) [110]. There are currently several FDA-approved liquid biopsy tests available for therapeutic target detection, and in the United Kingdom, the Galleri trial is exploring the utility of cell-free DNA in early cancer detection in a large multi-centre study [111,112]. Many early detection approaches utilise high-dimension data, although are good candidates for enhancement with AI. As recent examples, Tao et al. developed a modified random forest algorithm to diagnose hepatocellular carcinoma using whole-genome data, achieving a maximum validation AUC of 0.920 [113], and a DL model to analyse the Raman spectroscopy of liquid biopsy blood exosomes had an AUC of 0.912 for lung cancer detection [114]. In the literature, a wide number of ML techniques have been applied to liquid biopsy material, including linear models, support vector machines, decision trees, and deep learning models, with excellent AUCs for cancer detection [115,116,117,118]. CancerSEEK is a notable example: the test can detect eight common cancer types through analysis of cell-free DNA, and is based on a random forest model evaluating eight proteins and 1933 gene positions [118]. CancerSEEK can predict malignancy with an AUC of 91%, and although performance varied across tumour groups, it identified a very high proportion of ovarian and liver cancers [118]. It is likely that ML-enhanced methods will play a central role in high-dimensional cancer biomarker analysis, particularly as the amount of extractable data increases and the appetite to combine imaging with liquid biopsy and digital pathology data evolves [119]. Anticipating this fact, and acknowledging possible barriers to entry, Issadore and colleagues have developed a user guide for applying ML to liquid biopsy data, as well as a web-based tool which automates model generation without user input [120,121]. 

### 3.5. Early Detection of Recurrence

Another application of AI to oncology which is making strides is improved prognostication and earlier recurrence detection following treatment. In the pre-treatment setting, accurate prognostication could facilitate personalised therapy [122], so that cases identified as high-risk may be offered more intensive primary treatment, for example, radiotherapy dose escalation, whereas lower risk patients could be stratified to less intensive treatment to reduce side effects [123,124]. 

Post-treatment surveillance is a universally recommended aspect of cancer care, which offers patients ongoing support for treatment-related side effects, reassurance and management of co-morbidities [125]. Increased surveillance intensity according to risk could facilitate earlier treatment for recurrence, or improve early diagnoses of second primary cancers, especially where shared risk factors exist [126,127]. In addition, stratified surveillance may enable optimal resource allocation and have significant financial benefits [128]. ML using routinely available clinical data (patient, tumour and treatment characteristics) has been able to predict recurrence of bladder cancer at 1, 3, and 5 years post-cystectomy with greater than 70% sensitivity and specificity [129]. Such models provide a useful prognostic benchmark in a tumour that otherwise lacks widely recognised biomarkers.

Digital pathology has also been utilised for ML-based recurrence prediction, and has shown promise for several cancers including hepatocellular carcinoma (HCC), bladder, melanoma and rectal cancers [130,131,132,133]. Yamashita et al. developed a deep learning model for recurrence risk following the surgical resection of HCC, with performance exceeding TNM-based prognostication [131]. The model successfully stratified patients into high- and low-risk groups with statistically significant survival differences [131]. Jones et al. discovered that the ratio of desmoplastic to inflamed stroma predicts disease recurrence in locally excised rectal cancer [133]. This novel marker can be assessed on a single H&E section, thus offering easily accessible information to guide further management [133].

The number of imaging-based radiomic and DL models for the prognostication of post-treatment recurrence has grown considerably over the last decade [134]. Shen et al. used deep learning with PET scans to develop a model to predict local recurrence of cervical cancer following chemoradiotherapy. Test set sensitivity and specificity were 71% and 93%, respectively; however, the study was limited by a small, single-centre sample size [135]. Zhang et al. applied machine learning to pre-operative CT-derived radiomic and clinical features to develop a recurrence prediction model for gastric cancer. With an external test set AUC of 0.808 (confidence interval 0.732–0.881), this model lays the foundation for future pre-operative personalised prognostic tools to guide further treatment in gastric cancer [136].

DL combined with radiomics has been used to predict treatment failure following stereotactic ablative radiotherapy (SABR) in NSCLC and make recommendations towards individualised radiotherapy doses to reduce failure-risk [137]. When combined with clinical features, the ‘Deep Profiler’ model had a concordance index of 0.72 (95% CI 0.67–0.77) for predicting local treatment failure. Results from this study suggest the existence of image-distinct subpopulations with varying sensitivity to radiation, and that AI can be used to individualise radiotherapy doses [137].

## 4. Challenges and Future Directions

The promise of healthcare AI comes with several challenges, including ethical considerations, algorithmic fairness, data bias, governance and security [138,139,140] (Figure 6). Developing ethical principles and frameworks is the subject of significant ongoing work in healthcare AI [141]. The WHO have called on healthcare AI stakeholders to ensure that new technologies place ethics and human rights at the centre of their design and use [142]. Although a detailed analysis of ethical issues is beyond the scope of this review, we have previously discussed common concerns, including the black-box nature of AI decisions, the impact on patient experience and shared decision-making, and where responsibility lies if AI fails to make accurate predictions [143].

As the field evolves, there is increasing awareness of the negative consequences of model bias, particularly in respect to demographic characteristics such as sex and ethnicity. As one example, an AI-tool for diagnosing skin cancer based on 129,450 clinical images achieved parity with dermatologists [144], but less than 5% of images pertained to darker skin, drawing criticism about reproducibility and external validity [145]. A large meta-analysis published recently concluded that ethnicity data are available for only 1.3% of images in publicly available skin datasets, with ‘substantial underrepresentation of darker skin types’ [146]. A commentary by Robinson et al. highlights that understanding and addressing structural racism and bias is likely to improve both model accuracy and external validity, and we hope that measures to describe ethnic distributions and address biases will become increasingly adopted [147]. 

Data curation and storage can be time consuming and costly, and with the increasing focus on data stewardship amongst the scientific community, many agencies now require clear plans for data management from the outset [148]. The FAIR guiding principles for data management aim to assist researchers in good data stewardship, as well as in maximising the utility of datasets to the broader research community [148]. 

The requirement for large sets of labelled data for model training, which are time consuming and costly to generate, presents a significant challenge to researchers. Methods to circumvent data limitations have historically included transfer learning, whereby models pre-trained on larger datasets, such as ImageNet, are applied to a new problem [149]. However, as the field evolves sophisticated solutions are arising, including self-supervised learning, whereby visual representations of unannotated data are used to assign labels based on similarity measures [150,151]. A recent Google paper showed that this approach had better image classification performance than traditional labelled methods [152]. Many groups are also utilising synthetic data to boost sample sizes. For example, Liu et al. used a generative adversarial network (GAN) to create synthetic serum glycosylation data, leading to improvements in hepatocellular carcinoma diagnosis and staging [153]. 

Data security is also an ongoing concern, especially in light of recent high-profile leaks and the potential threat of inference attacks [154,155]. Approaches are emerging to improve data security and reduce the risks associated with transferring data across multiple institutions. In 2016, Google introduced the term ‘Federated Learning’, referring to the process of training models peripherally without movement of sensitive data to the central institution [156]. Kaissis et al. recently released PriMIA (privacy-preserving medical image analysis), an open-source framework to enable federated medical imaging analysis of encrypted data across institutions [157]. The process works by sending the untrained model from the central server, training it locally at each institution, and periodically aggregating the results centrally for inference. The authors deployed a federated DL model for paediatric chest X-ray classification across three institutions, which performed as well as non-secure local models and was robust to model inversion attacks [157]. Such approaches may help to allay data sharing concerns in the future. 

Perhaps the most significant criticism of AI is that many models have not been evaluated with the same rigour as expected for other medical interventions. Firstly, as mentioned above, some tools have been used clinically without peer-reviewed publication, meaning they have not been subjected to the standard of rigorous adversarial feedback expected by the scientific community. Moreover, if the methodology is not published it cannot be reproduced, which is concerning given claims of an ongoing reproducibility crisis in academia [158]. In addition, despite the rapid rise in AI publications utilising extremely large datasets, there is a notable paucity of prospective studies [159]. The randomised control trial (RCT) has long been the gold standard for medical interventions, but again, these are highly uncommon for AI models [159]. This is a problem, because the few RCTs we do have suggest performance is likely to drop when evaluated under RCT conditions [160]. A recent systematic review of AI systems in breast cancer screening found that many were of poor methodological quality, and promising results from small studies did not carry over to larger trials [161]. Finally, many retrospective models are not externally validated, again leading to overly optimistic performance estimates [162]. Models which are not externally validated do not provide good evidence of generalisability required for clinical adoption. A number of frameworks have been developed to improve the standard of healthcare AI publications, including CONSORT-AI, SPIRIT-AI and TRIPOD-AI [163], as well as a toolkits to empower clinicians to critically appraise such studies [164].

## 5. Conclusions

We have seen that the application of AI to healthcare data has the potential to revolutionise early cancer diagnosis and provide support for capacity concerns through automation. AI may allow us to effectively analyse complex data from many modalities, including clinical text, genomic, metabolomic and radiomic data.

In this review, we have identified myriad CNN models that can detect early-stage cancers on scan or biopsy images with high accuracy, and some had a proven impact on workflow triage. Many commercial solutions for automated cancer detection are becoming available, and we are likely to see increasing adoption in the coming years. 

In the setting of symptomatic patient decision-support, we argue that caution is needed to ensure that models are validated and published in peer-reviewed journals before use. Moreover, we identified a number of challenges to the implementation of AI, including data anonymisation and storage, which can be time-consuming and costly for healthcare institutions. We also addressed model bias, including the under-reporting of important demographic information such as race and ethnicity, and the implications this can have on generalisability. 

In terms of how study quality and model uptake can be improved going forwards, quality assurance frameworks (such as SPIRIT-AI), and methods to standardise radiomic feature values across institutions, as proposed by the image biomarker standardisation initiative, may help [165]. Moreover, disease-specific, ‘gold standard’ test sets could help clinicians benchmark multiple competing models more readily. 

Despite the above challenges, the implications of AI for early cancer diagnosis are highly promising, and this field is likely to grow rapidly in the coming years. 

## Figures and Tables

**Figure 1 cancers-14-01524-f001:**
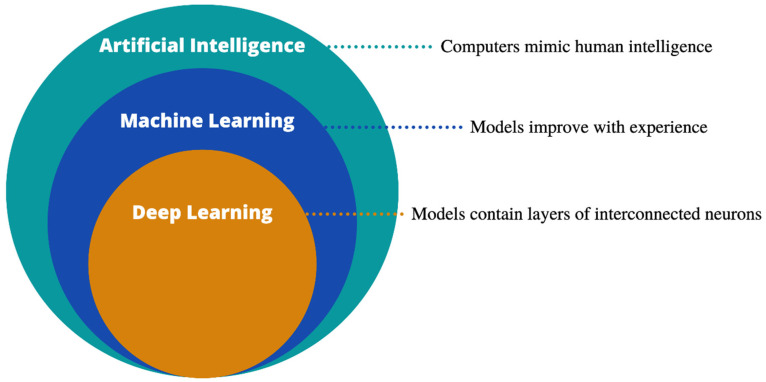
Artificial intelligence and its sub-divisions.

**Figure 2 cancers-14-01524-f002:**
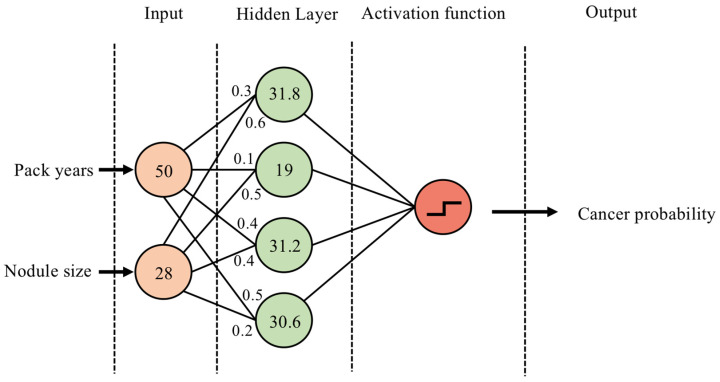
Example of a single-hidden-layer ANN architecture. (1) The smoking status in pack years and lung nodule size (mm) are entered as the two input nodes. (2) In the hidden layer, each node multiplies the values from incoming neurons by a weight (shown as decimals at incoming neurons) and aggregates them. (3) The results are passed to an activation function, converting the output to a probability of cancer between 0 and 1. Multiple learning cycles are used to update the hidden layer weights to improve performance.

**Figure 3 cancers-14-01524-f003:**
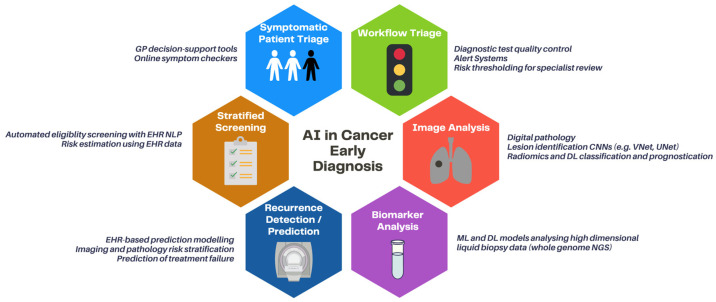
Clinical applications of AI in early cancer diagnosis. Abbreviations: GP: general practitioner, NLP: natural language processing, EHR: electronic healthcare record, ML: machine learning, DL: deep learning, NGS: next-generation sequencing.

**Figure 4 cancers-14-01524-f004:**
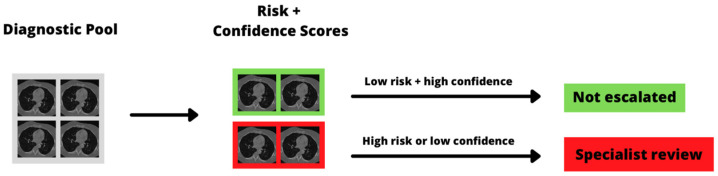
Example diagnostic triage pipeline. The AI model assigns a risk group to each examination, as well as a confidence estimate, and scans that are either high risk or have low diagnostic confidence are escalated for specialist review. CT images taken from the public LUNGx dataset [97].

**Figure 5 cancers-14-01524-f005:**
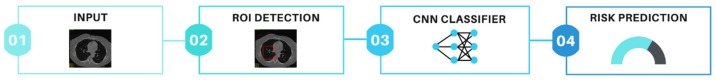
Example of an ‘end to end’ cancer detection pipeline. 1: A whole CT volume is used as input into the model. 2: A region detection architecture (such as UNet) is used to identify a sub-volume and assign a bounding-box ROI. 3: The volume encompassed by the ROI is input into a classification CNN (such as InceptionNet) to learn patterns associated with the outcome variable. 4: A risk prediction of malignancy is output. Abbreviations: ROI: region of interest, CNN: convolutional neural network. CT images taken from the public LUNGx dataset [97].

**Figure 6 cancers-14-01524-f006:**
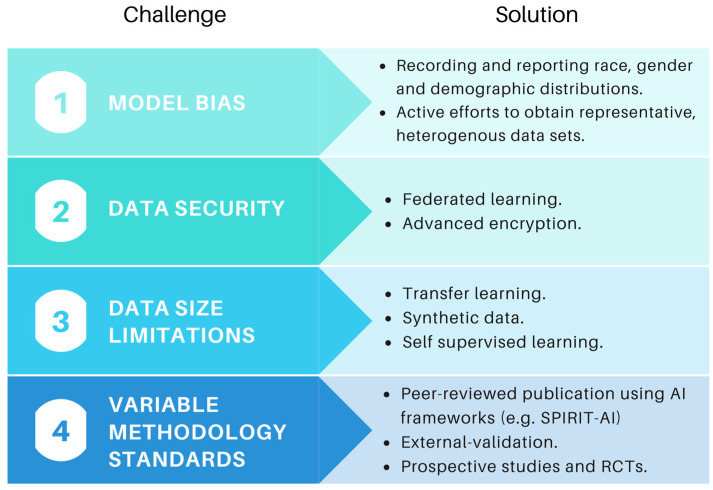
Challenges and possible solutions to improve the robustness of AI models in the future.

**Table 1 cancers-14-01524-t001:** Common supervised ML techniques with early diagnosis examples.

Model	Type	Description	Example
LR	R	Uses logistic function to predict categorical outcomes	Chhatwal et al. [13]
SVM	R, C	Constructs hyperplanes to maximise data separation	Zhang et al. [14]
NB	C	Utilises Bayesian probability including priors for classification	Olatunji et al. [15]
RF	R, C	Ensembles predictions of random decision trees	Xiao et al. [16]
XGB	R, C	As RF, but sequential errors minimised by gradient descent	Liew et al. [17]
ANN	R, C	Multiplies input by weights and biases to predict outcome	Muhammad [18]
CNN	R, C	Uses kernels to detect image features	Suh [19]

Abbreviations: R: regression, C: classification, LR: logistic regression, SVM: support vector machine, NB: naïve Bayes, RF: random forest, XGB: extreme gradient boosting, ANN: artificial neural network, CNN: convolutional neural network.

**Table 2 cancers-14-01524-t002:** Possible benefits and limitations of traditional ML vs. deep learning.

Traditional Machine Learning	Deep Learning
Requires ROI segmentation	ROI segmentation optional
Features are pre-specified	Features generated by model
Features are easily quantified	Features difficult to quantify
Computationally less intensive	Computationally more intensive
May perform better on small datasets	May perform better on large datasets

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
