# Peer review of "The Role of Artificial Intelligence in Early Cancer Diagnosis"

_cancers, 2022, doi:10.3390/cancers14061524_

Round 1
Reviewer 1 Report
This is a well structured and concise review of the highly actual field of AI in medicine. The examples are mainly, but not only taken from the field of lung disease according to the field the authors are working in. The review contains a basic description of modern techniques of artificial intelligence allowing the not especially involved clinician an insight into these techniques without overwhelming him with technical details not needed to understand the promises and possible pitfalls that are described for a whole range of applications discussed. Besides the promising results shown with in part very large numbers of patients also the shortage of prospective studies has been rightly pointed out.
Only a few formal edits must be noted:
Line 14: early diagnosis does increase the chance of effective treatment in many, but not in all cancers
Line 22: World Health Organisation is to be written with capital letters
Line 51: The sentence should end with a point instead of a comma after mortality.
Line 171: “DICE” score should be explained the first time it is used. The term is not yet familiar to the average clinician
Line 184: “ED” should be explained
Line 402: the references 127-130 deal with melanoma, HCC, bladder and rectal cancer, not breast cancer
The literature coverage is adequate. I only missed one important recent analysis highlighting the possibility of overestimating the quality of AI diagnosis for screening mammography:
Freeman K, Geppert J, Stinton C, Todkill D, Johnson S, Clarke A, Taylor- Phillips S. Use of artificial intelligence for image analysis in breast cancer screening programmes: systematic review of test accuracy. BMJ. 2021 Sep 1;374:n1872. doi: 10.1136/bmj.n1872.
Also Ref. 20 is incomplete and should be completed
Author Response
Dear reviewer,
Thank you very much for your comments, which we agree strengthen the manuscript. Please see our responses to your points below. We have uploaded the corrected manuscript with changes highlighted in yellow, and referenced the corresponding pages in our responses.
Line 14: early diagnosis does increase the chance of effective treatment in many, but not in all cancers.
Thank you. This has now been changed to read: "Diagnosing cancer at an early stage increases the chance of effective treatment in many tumour groups." (Page 2 of the attached document).
Line 22: World Health Organisation is to be written with capital letters
This has now been capitalised (Page 2)
Line 51: The sentence should end with a point instead of a comma after mortality.
Thank you. We have now changed the wording of this sentence as follows: "Many studies indicate screening can improve early cancer detection and mortality, but even in disease groups with established screening programmes such as breast cancer, there are on-going debates surrounding patient-selection and risk-benefit trade-offs, and concerns have been raised about a perceived ‘one size fits all’ approach incongruous with the aims of personalised medicine." (Page 3)
Line 171: “DICE” score should be explained the first time it is used. The term is not yet familiar to the average clinician
We have now added the following explanatory sentence: "Such models can be evaluated using the Dice similarity co-efficient (dice-score), which assesses the degree of overlap between two segmentation masks." (Page 8)
Line 184: “ED” should be explained
"ED" has been changed to "early diagnosis" (Page 8)
Line 402: the references 127-130 deal with melanoma, HCC, bladder and rectal cancer, not breast cancer
Thank you for highlighting this mistake. Breast cancer has been removed, and the sentence now reads: "Digital pathology has also been utilised for ML-based recurrence prediction, and has shown promise for several cancers including hepatocellular carcinoma (HCC), bladder, melanoma and rectal cancers" (Page 16)
The literature coverage is adequate. I only missed one important recent analysis highlighting the possibility of overestimating the quality of AI diagnosis for screening mammography:
Freeman K, Geppert J, Stinton C, Todkill D, Johnson S, Clarke A, Taylor- Phillips S. Use of artificial intelligence for image analysis in breast cancer screening programmes: systematic review of test accuracy. BMJ. 2021 Sep 1;374:n1872. doi: 10.1136/bmj.n1872.
Thank you for highlighting this notable paper, which has now been referenced in the challenges section: "A recent systematic review of AI systems in breast cancer screening found that many were of poor methodological quality, and promising results from small studies did not carry over to larger trials[163]" (Page 19-20)
Also Ref. 20 is incomplete and should be completed
Thank you for highlighting this error. This reference is now completed and reads:
"Szegedy, C.; Liu, W.; Jia, Y.; Sermanet, P.; Reed, S.; Anguelov, D.; Erhan, D.; Vanhoucke, V.; Rabinovich, A. Going deeper with convolutions. In Proceedings of the Proceedings of the IEEE Computer Society Conference on Computer Vision and Pattern Recognition; 2015; Vol. 07-12-June, pp. 1–9." (Ref 22, page 24)

Reviewer 2 Report
The present paper is a well written comprehensive review on the potential applications of AI for early cancer diagnosis 61 in symptomatic and asymptomatic patients, focusing on the types of data that can be used and the clinical areas most likely to see impact in the near future. Emphasis is also given to several challenges (algorithmic fairness, data bias, governance and security) and possible solutions to improve the robustness of AI models and their external validity. Authors conclude that limitations exists and that caution is needed to ensure emerging technologies are prospectively evaluated, externally validated and published in peer-reviewed journals to ensure patient safety and transparency.
Overall, I find this a well-timed article, as AI models are expected to gain popularity in the field of precision medicine. The paper is scientifically accurate and fully comprehensive to the reader. The reference section is adequate, up-to-date and appropriate to back up the points made by the Authors. I have no major comment on this well presented review.
Author Response
Dear reviewer,
Thank you very much for your comments on the manuscript. In response to the other reviewers' suggestions, we have uploaded an updated version of the manuscript below, with changes highlighted in yellow.

Reviewer 3 Report
The review is very informative and written with clear logic. However, I also have major concerns that should be addressed by the authors in the attempt to improve the overall quality.
- Cancer is a complex disease and to understand its complexity, it is imperative to take an integrative and holistic approach that combines multi-omics data to dissect and understand the interrelationships of the involved molecules. It also helps in assessing the flow of information from one omics to the other and helps in bridging the gap among them. I would suggest author to incorporate a section and discuss the integration of multi-omics data in cancer diagnosis and also discuss the importance of integrating multi-omics data over single omics data analysis and how these multi-omics approaches can improve the diagnosis process.
- It would be useful to report current ML-based approaches in oncology by considering single cancer type.
- Author should also comment the applicability of ML-based algorithms in cancer diagnosis for large scale use in terms of Public Health.
- In conclusion section, author conclude it in very brief, I would recommend author to discuss it in detail.
Author Response
Dear reviewer,
Thank you very much for your comments on the manuscript. Please see our responses to your points below. We have uploaded the corrected manuscript with changes highlighted in yellow, and reference the corresponding pages in our responses.
1) Cancer is a complex disease and to understand its complexity, it is imperative to take an integrative and holistic approach that combines multi-omics data to dissect and understand the interrelationships of the involved molecules. It also helps in assessing the flow of information from one omics to the other and helps in bridging the gap among them. I would suggest author to incorporate a section and discuss the integration of multi-omics data in cancer diagnosis and also discuss the importance of integrating multi-omics data over single omics data analysis and how these multi-omics approaches can improve the diagnosis process.
Thank you for highlighting the importance of multi-omics approaches to early cancer diagnosis. We have now added a 'multi-omics' section on pages 9-10. Here we discuss how multi-omics data including RNA sequencing and methylation data have been used to discover subclassess of glioblastoma, and mention how such approaches are important to fully understand tumour biology.
"Data Types: Multi-omic Data
Given the complexity of tumour biology, models based on single data types could miss important predictive information arising from the interaction between interdependent biological systems. There is therefore a drive to integrate multi-model data, which may include radiomic, genomic, transcriptomic, metabolomic and clinical factors, to better describe the tumour landscape and improve diagnostic precision. Several large-scale databases, including ‘LinkedOmics’, which contains multi-omic data for 11,158 patients across 32 cancer types, are available to facilitate detection of associations between data modalities and assist model development[76].
Using central nervous system (CNS) tumours as an example, multi-omic data, including single-nucleotide polymorphism (SNP) mutations (e.g. TARDBP), gene methylation (e.g. 64-MMP) and transcriptome abnormalities (e.g. miRNA-21), are known to predict progression of meningiomas [77]. A systematic review of multi-omic glioblastoma studies by Takahashi et al found that most utilised ML techniques for analysis, likely due to the size and complexity of the data[78]. In one study of 156 patients with oligodendrogliomas, mRNA expression arrays, microRNA sequencing and DNA methylation arrays were analysed using a multi-omics approach to better classify 1p/19q co-deleted tumours [79]. Use of unsupervised clustering techniques identified previously undescribed molecular heterogeneity in this group, revealing three distinct subgroups of patients [79]. These subgroups had differences in important histological factors (microvascular proliferation and necrosis), genetic factors (cell-cycle gene mutations) and clinical factors (age and survival)[79]. Franco et al explored DL autoencoder models to predict cancer subtypes from multi-omic data, included methylation, RNA and miRNA sequencing readouts from The Cancer Genome Atlas (TCGA)[80]. The authors identified three GBM subtypes with differentially expressed genes relating to synaptic function and vesicle-mediated transport[80].
These studies demonstrate how machine learning approaches applied to multi-omic data can reveal previously hidden elements of tumour biology, which may have important implications for diagnosis and prognostication."
2) It would be useful to report current ML-based approaches in oncology by considering single cancer type.
Thank you for this comment. We considered this approach when originally proposing the article to the special edition editors, and decided that it would be best to give an overview of multiple cancer types rather than focussing on a single subgroup. In our opinion we have covered the relevant tumour groups where exciting results have been found, and believe this broader approach will remain interesting for a general oncology readership.
3) Author should also comment the applicability of ML-based algorithms in cancer diagnosis for large scale use in terms of Public Health.
Thank you. We agree it is important to consider the implications of AI on cancer early diagnosis in a wider public health setting. In section 3.1. (Page 11) on risk-stratified screening of asymptomatic patients, we cover a study by Gould which included over 190,000 patients in the normal population. The model was able to predict the risk of lung cancer in the following 12 months, and was used to identify asymptomatic members of the population who would be suitable for lung cancer screening. We feel that this section gives an example of how AI can be used on a population level to predict cancer risk.
4) In conclusion section, author conclude it in very brief, I would recommend author to discuss it in detail.
Thank you for your feedback on the conclusion section. This has now been expanded to give more detail (page 20).
"Conclusion
We have seen that the application of AI to healthcare data has the potential to revolutionise early cancer diagnosis and provide support for capacity concerns through automation. AI may allow us to effectively analyse complex data from many modalities, including clinical text, genomic, metabolomic and radiomic data.
In this review we identified myriad CNN models that can detect early-stage cancers on scan or biopsy images with high accuracy, and some had a proven impact on workflow triage. Many commercial solutions for automated cancer detection are becoming available, and we are likely to see increasing adoption in the coming years.
In the setting of symptomatic patient decision-support, we argue that caution is needed to ensure models are validated and published in peer-reviewed journals before use. Moreover, we identified a number of challenges to implementation of AI, including data anonymisation and storage, which can be time-consuming and costly for healthcare institutions. We also addressed model bias, including the under-reporting of important demographic information such as race and ethnicity, and the implications this can have on generalisability.
In terms of how study quality and model uptake can be improved going forwards, quality assurance frameworks (such as SPIRIT-AI), and methods to standardise radiomic feature values across institutions, as proposed by the image biomarker standardisation initiative, may help[167]. Moreover, disease-specific, ‘gold standard’ test sets could help clinicians benchmark multiple competing models more readily.
Despite the above challenges, the implications of AI for cancer early diagnosis are highly promising, and this field is likely to grow rapidly in the coming years."

Reviewer 4 Report
The authors give an extensive review of artificial intelligence in cancer early diagnosis. The paper in general is well written, and I tend to support the publication of this paper. Some comments/suggestions are shown below for the improvement of the paper.
- As the authors have pointed out, deep learning is a subgroup of machine learning. However, in Table 1, deep learning was not included, which is a pity. In my mind, the highlight of this paper is to review forefront machine leaning techniques, and I think that deep learning (and the other relevant approaches) should be included in Table 1.
2. Section 2.2 seems to talk about the use of NLP in EHR data. However, NLP was never mentioned in previous sections, so the readers are not well prepared when NLP suddenly becomes a big topic here. It might be helpful to mention NLP somewhere earlier in the paper to make the paper more coherent.
3. In Section 2.3, the authors seem to be treating 'radiomics' and 'deep learning' as two categories of analysis approaches. I am not sure that such a categorization is widely accepted. In my mind, radiomics just refers to the radio-imaging data, not a type of analysis approach. In fact, the cited example (Lu et al, 2019) used lasso to reduce 42 features to 4 features, where lasso is a machine learning approach. So, I think the authors are actually comparing 'simple machine learning approaches' vs 'DL', rather than 'radiomics' vs 'DL'.
4. I agree with the authors that external-validation and prospective studies and RCTs are needed to better assess the accuracy of emerging AI models in early cancer diagnosis. Would the authors suggest some practical strategies to support such ideas? For example, advocate to establish a 'gold-standard testing dataset' so that various AI models can be tested and compared? Or advocate to set up a website to compare various studies/approaches and point out their pros and cons? Without such effort, people will soon be inundated by a sea of AI models, without knowing which one should be really adopted in practice.
Author Response
Dear reviewer,
Thank you very much for your comments, which we agree strengthen the manuscript. Please see our responses to your points below. We have uploaded the corrected manuscript with changes highlighted in yellow, and reference the corresponding pages in our responses here.
1) As the authors have pointed out, deep learning is a subgroup of machine learning. However, in Table 1, deep learning was not included, which is a pity. In my mind, the highlight of this paper is to review forefront machine leaning techniques, and I think that deep learning (and the other relevant approaches) should be included in Table 1.
Thank you for your feedback. Table 1 (page 4) has now been expanded to include deep-learning models, with example early diagnosis publications.
2. Section 2.2 seems to talk about the use of NLP in EHR data. However, NLP was never mentioned in previous sections, so the readers are not well prepared when NLP suddenly becomes a big topic here. It might be helpful to mention NLP somewhere earlier in the paper to make the paper more coherent.
Thank you for pointing this out. The concept of NLP is now introduced earlier in the text to improve coherence. The following line has been moved to the 'definitions' section on page 4:
"When analysing unstructured clinical data, an often utilised technique, both in oncology and more broadly, is Natural Language Processing (NLP)[12]. NLP transforms unstructured free-text into a computer-analysable format, allowing automation of resource-intensive tasks."
The section on use of NLP for EHR data (Page 6) now reads: "EHR data typically include structured, easily quantifiable data such as admission dates or blood results, and unstructured free-text such as clinical notes or diagnostic reports. The latter can be analysed using NLP approaches."
3. In Section 2.3, the authors seem to be treating 'radiomics' and 'deep learning' as two categories of analysis approaches. I am not sure that such a categorization is widely accepted. In my mind, radiomics just refers to the radio-imaging data, not a type of analysis approach. In fact, the cited example (Lu et al, 2019) used lasso to reduce 42 features to 4 features, where lasso is a machine learning approach. So, I think the authors are actually comparing 'simple machine learning approaches' vs 'DL', rather than 'radiomics' vs 'DL'.
Thank you for highlighting this issue. We have now corrected the text so that radiomics refers only to the radio-imaging data, and the comparisons are between 'traditional machine learning' and 'deep learning' methods. Please see the corrections to Table 2 and the explanatory text below starting on Page 7.
4. I agree with the authors that external-validation and prospective studies and RCTs are needed to better assess the accuracy of emerging AI models in early cancer diagnosis. Would the authors suggest some practical strategies to support such ideas? For example, advocate to establish a 'gold-standard testing dataset' so that various AI models can be tested and compared? Or advocate to set up a website to compare various studies/approaches and point out their pros and cons? Without such effort, people will soon be inundated by a sea of AI models, without knowing which one should be really adopted in practice.
Thank you for these interesting ideas, which we have addressed in the expanded conclusions section (page 20). As you say, gold standard test sets and methods to easily compare model performance are definitely needed, and may make clinical adoption easier:
"In terms of how study quality and model uptake can be improved going forwards, quality assurance frameworks (such as SPIRIT-AI), and methods to standardise radiomic feature values across institutions, as proposed by the image biomarker standardisation initiative, may help[167]. Moreover, disease-specific, ‘gold standard’ test sets could help clinicians benchmark multiple competing models more readily. "

Round 2
Reviewer 3 Report
This is a highly informative review. Authors addressed all the comments raised by me. I would recommend it for the acceptance.
Author Response
Thank you for your comments.